



**The miscellaneous synoptic forcings in the four-day widespread extreme rainfall**
**event over North China in July 2023**
Jinfang YIN[1,2,3], Feng LI[1], Mingxin LI[1], Rudi XIA[1], Xinghua BAO[1],
Jisong SUN[1], and Xudong LIANG[1]
[1] State Key Laboratory of Severe Weather, Chinese Academy of Meteorological
Sciences, Beijing 100081, China
[2] Research Center for Disastrous Weather over Hengduan Mountains & Low-Latitude
Plateau, China Meteorological Administration (CMA), Kunming 650034, China
[3] Shigatse National Climatological Observatory, CMA, Shigatse 857000, China
Submitted to *Natural Hazards and Earth System Sciences* (*NHESS*)
August 2024
Corresponding author: Jinfang YIN
E-mail: yinjf@cma.gov.cn




**ABSTRACT**

Synoptic forcings have traditionally played a pivotal role in extreme rainfall over North
China. However, there are still large unexplained gaps in understanding the formation of
extreme rainfalls over this region. The heavy rainfall event, lasting from 29 July to 2 August
2023 ( referred to as "23·7" event), is characterized by long duration, widespread coverage,
and high accumulated rainfall over North China. Overall, the persistent extreme rainfall is
closely associated with the remnant vortex originating from typhoon Doksuri(2305), tropical
storm Khanun(2306), and the unusual westward extended western Pacific subtropical high
(WPSH), as well as quasi-stationary cold dry air masses surrounding North China on the west
and north sides. Based on wind profiles and rainfall characteristics, the life history of the
"23·7" event is divided into two stages. In the first stage, the western boundary of the western
Pacific subtropical high (WPSH) was destroyed by the tropical storm Doksuri, appearing that
the WPSH retreated eastward with decreasing height. As a result, an inclined vertical
distribution on the western boundary was established below 500 hPa. Therefore, convections
were limited by the tilted WPSH with warm-dry cover embedded in the low-to-middle
troposphere. Meanwhile, the orography in the west of North China was controlled by cold air
masses above nearly 3.0 km. Combining the orographic and cold air blockings, only a shallow
southeasterly layer (between 1.3 and 3.0 km) can overpass mountains. Although the warm
and moist southeasterly flows were lifted by orography, no convections were triggered
because of the local capped cold and dry air masses overhead. Under this framework,
equivalent potential temperature ($\theta_e$) gradients were established between warm humid and
dry cold air masses, similar to a warm front, causing warm air to lift and generate widespread
rainfall but low intensity. However, the lifting was too weak to allow convection to be highly
organized. In the second stage, the WPSH was further destroyed by enhanced Khanun, and
thus the embedded warm-dry cover associated with the tilted WPSH was significantly thinned.
Consequently, convections triggered by orographic blocking can move upward and
consequently further develop, forming deep convections. Comparatively speaking, the
convections in the second stage are much deeper than those in the first stage. The results
gained herein may shed new light on better understanding and forecasting of long-lasting
extreme rainfall.


## 1. Introduction

A persistent severe rainfall event occurred over central and North China during the period from 29 July to 2 August 2023, which was regarded as one of the precipitation extremes of 2023 globally (Fowler et al., 2024). Despite the rainfall in low intensity, it was long-lasting and widespread, resulting in large accumulated rainfall. Flooding from this event affected 1.3 million people, bringing severe human casualties and economic losses. One of the distinct features of this rainfall event was closely associated with the remnant vortex originating from typhoon Doksuri(2305), tropical storm Khanun(2306), unusual westward extended western Pacific subtropical high (WPSH), and quasi-stationary cold dry air masses surrounding North China on the west and north sides.

It is common for rainfall occurrence over North China due to strong water vapor supply by tropical cyclones over the East China Sea and/or Southern China Sea (e.g., Ding,1978; Feng and Cheng,2002; Yin et al.,2022c). Like the "96·8" heavy rainfall event (Sun et al.,2006; Bao et al., 2024), the present persistent rainfall event was closely linked to two tropical storms of Doksuri and Khanun. Note that the Doksuri weakened to a typhoon remnant vortex (typhoon-low pressure) at this moment as it moved inland after landfalling, while the tropical storm Khanun was in a fast-developing stage. The tropical storm Khanun and the typhoon remnant vortex built a water vapor bridge, transporting a large amount of water vapor to North China from the East China Sea. Previous studies (e.g., Hirata and Kawamura,2014; Gao et al.,2022; Yang et al.,2017) pointed out that such large-scale weather conditions were favorable for heavy rainfall generation.

In the last several decades, considerable attention was paid to the remote rainfall events associated with tropical cyclones, and substantial progress has been made ( e.g., Wang et al., 2009; Xu et al.,2023a; Xu et al.,2023b; Lin and Wu,2021). Commonly, sufficient water vapor provided by a tropical cyclone plays an important role in extreme rainfall over North China (e.g., Rao et al.,2023; Xu et al.,2023b). Besides, many studies confirmed that the WPSH is closely related to water vapor transportation and the spatial distribution of surface rainfall (e.g., Hu et al.,2019; Gao et al.,2022). Additionally, orographic forcing of the approaching warm and moist unstable air plays a critical role in determining the location of convection initialization,




although sometimes orographic forcing played a small role compared to Typhoon's circulation
(Wang et al.,2009). Moreover, heavy rainfall can be generated by the complicated cloud
microphysical processes due to the interactions between tropical oceanic warm-moist and mid-
latitude cold-dry air masses (Wang et al.,2009; Xu and Li,2017; Xu et al.,2021). Despite some
experiences gained, there are still large unexplained gaps in understanding the formation of
extreme rainfall (Meng et al.,2019). In this event, no highly organized strong convective system
was observed, and rainfall was featured by long duration, widespread coverage, and high
accumulation. Although operational forecasts gave reasonable results at that time, several
unique features emerged in this precipitation. Some unexplainable questions have been raised
after the persistent heavy rainfall event: (1) What mechanism(s) could account for the persistent
heavy rainfall? (2) What is the role of the unusual westward extended WPSH in governing the
rainfall over North China? Therefore, we are motivated to conduct the present modeling study
to answer those questions.

The rest of the paper is organized as follows. A detailed description of the main features

of extreme rainfall and synoptic-scale weather conditions is documented. Section 3 provides
detailed model configuration and verification against observations. We present a detailed
analysis of the extreme rainfall production in Section 4. The paper finishes with conclusions
and outlooks.
**2. Properties of rainfall and wind profiles**
*2.1 Characteristics of rainfall*

Figure 1 shows the spatial distribution of 96-h accumulated rainfall from observations

during the period from 0000 UTC 29 July to 0000 UTC 2 August 2023, with the peak amount
of 1004 mm at Liangjiazhuang station near Xingtai, Hebei Province of North China.
Exceptionally long duration of rainfall is a notable feature of the event, with the longest duration
being 80 hours within the four days at some stations. The spatial distribution of heavy rainfalls
is consistent with the orography of the Yanshan Mountains on the north and the Taihang
Mountains on the south, suggesting that the heavy rainfall may be associated with the orography.
It should be emphasized that three rainfall cores, marked by Mentougou (MTG) in Beijing, and
Yixian (YX) and Xiangtai (XT) in Hebei Province, correspond to the regions with large
topographic gradients (Fig. 1). Please refer to Li et al. (2024) for a detailed analysis of rainfall
fine features.

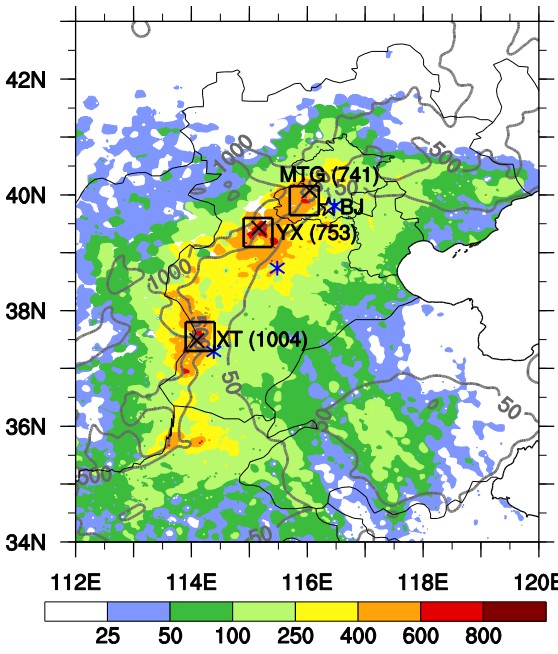


**Fig. 1** Spatial distribution of 96-h accumulated rainfall (mm, shadings) from the intensive

surface rain gauge observations during the period from 0000 UTC 29 July to 0000 UTC 2
August 2023; Gray contours denote orography from 50 m to 1000 m. Three rainfall cores in
Mentougou (MTG) in Beijing, and Yixian (YX) and Xiangtai (XT) in Hebei Province are
marked by squares, and the values in parentheses indicate the maximum accumulated rainfall
(marked by crisscross sign ×) for the regions, respectively. The blue asterisks (∗) represent the
locations of wind profiler observational stations near the three rainfall cores. The start (☆) sign
indicates the location of Beijing (BJ) City. (Similarly for the rest of figures).
***2.2 Wind profiles***

The observed wind profiles near MTG, YX, and XT are shown in Fig. 2. Obvious temporal

variations in horizontal wind fields can be seen during the rainfall event. Taking the wind
profiles near MTG as an example (Fig. 2a), the easterly or southeasterly wind at the levels
below 4 km gradually increased from 2 m s$^{-1}$ at 1200 UTC 28 to 24 m s$^{-1}$ at 1200 UTC 30 July
2023. The easterly or southeasterly wind lasted to 0400 UTC 31 July 2023, turned southerly
except for near the ground, and then turned southwesterly near 0400 UTC 1 August 2023. After
0400 UTC 31 July, wind speed decreased significantly and then increased drastically. More


specifically, the wind speed decreased from 8 m s⁻¹ to 2 m s⁻¹, then increased to 14 m s⁻¹ near 1
km above the ground. However, opposite variations can also be seen above 4 km. One can see
that the horizontal wind shifted from southwesterly to southerly, then back to southwesterly.
Overall, the shift in wind direction and speed altered vertical wind shear, which directly affected
the development and organization of subsequent convection (Pucik et al., 2021). Similar
variations can also be found at YX and XT stations, although the timing of changes is not
synchronized (Fig. 2c,d). The variations proceeded from south to north, starting first at XT and
finally at MTG, as the typhoon moved from south to north.

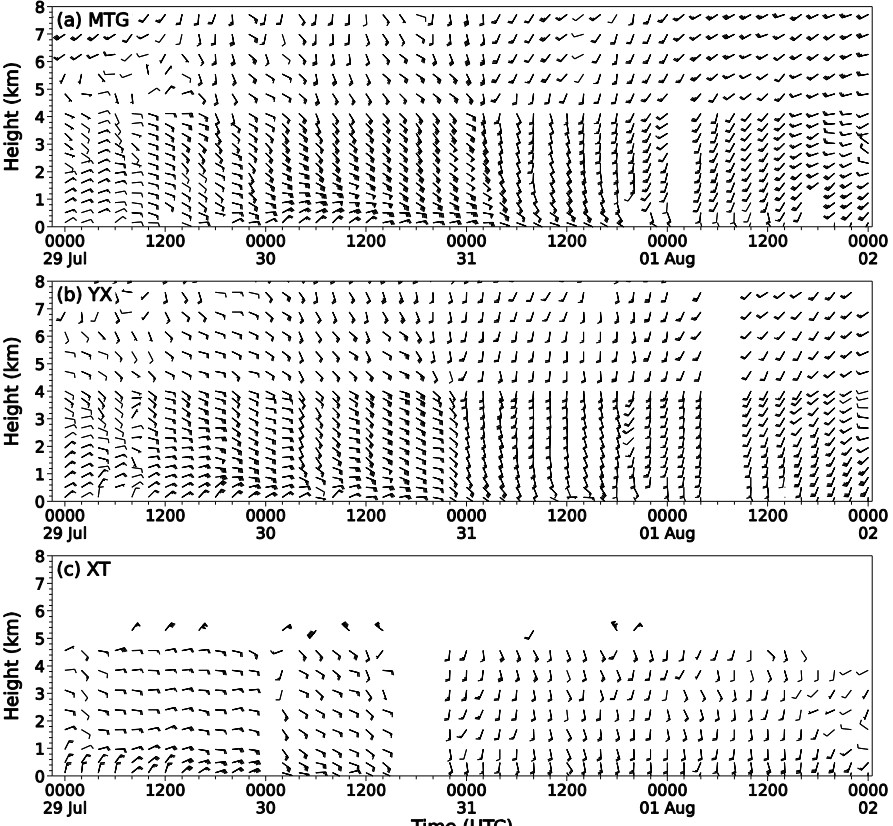


**Fig. 2** Temporal evolution of wind profile (a full barb is 4 m s⁻¹) from observations near (a)
MTG, (b) YX, and (c) XT during the period of 0000 UTC 29 July to 0000 UTC 2 August 2023.
Note only the wind profile below 5 km above the ground can be observed due to the limitation
of the instrumentation near Xingtai (XT). (see Fig. 1 their locations).


### 2.3 Synoptic conditions on 28 July 2023

Figure 3 displays a weather chart at 500 hPa at 1200 UTC 28 July 2023. One can see that the large-scale flow patterns exhibited a coexistence of a remnant vortex originating from typhoon Doksuri(2305)[*] and tropical storm Khanun(2306). The former weakened significantly into a vortex at this time, while the latter was in the rapid development stage. Another important weather system was the WPSH (denoted by the 588 isoline) with a square-head shape on its western border. Clearly, a water vapor transportation passage was built due to the cyclonic circulation of the tropical storm in combination with the anticyclonic circulation on the southwest of the WPSH. As a result, central and North China was covered by high precipitable water (PW) of over 68 mm. Similar patterns can be viewed at the level of 850 hPa (not shown).

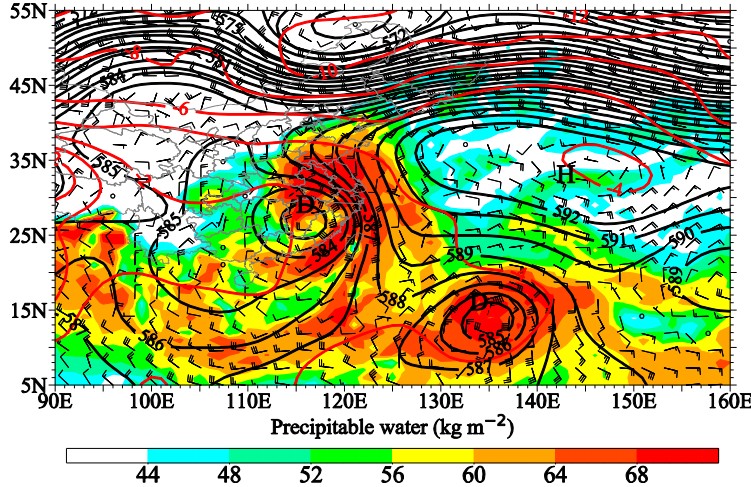

**Fig. 3** Weather chart at 500 hPa at 1200 UTC 28 July 2023: Geopotential height (black-contoured at 15 gpm intervals), temperature (red-contoured at 2°C intervals), wind barbs (a full barb is 4 m s[-1]), and precipitable water (kg m[-2], shadings).

### 3. Model configuration and verification

### 3.1 Model description

In this study, the persistent heavy rainfall event is reduplicated with the WRF model version 4.1.3. The WRF model is configured in two-way nested grids of horizontal grid sizes of 9 km, 3 km, and 1 km. Figure 4 displays the geographical coverage of the WRF model domains, with the grid points of 901(nx)×601(ny), 973×1231, and 1231×1591 for the outer,

---

[*]The typhoon Doksuri(2305) weakened to a typhoon remnant vortex as it was passing through East China's Anhui Province. The China Meteorological Administration (CMA) stopped issuing updates on the Doksuri at 0300 UTC 29 July 2023. The remnant of Doksuri remained in a vortex in the lower troposphere, although its wind force diminished as it moved northward.



intermediate, and inner domains, respectively. The outermost domain (i.e., D01) is centered at
115°E, 35°N, and a total of 58 sigma levels is assigned in the vertical with the model top fixed
at 20 hPa. Since the rainfall is closely related to the spatial distribution of orography over North
China (Fig. 1), the Shuttle Radar Topography Mission (SRTM) high-resolution (90 m)
topographic data is employed in the present simulation. It should be noted that the model
vertical level distribution was carefully tested and has achieved good performance (Yin et al.,
2020; Yin et al.,2022a; Yin et al.,2018; Yin et al.,2022b). The WRF model physics schemes are
configured with the YSU scheme for the planetary boundary layer (Hong et al., 2006), the
revised MM5 Monin-Obukhov (Jimenez) scheme for the surface layer (Jiménez et al., 2012),
and the Unified Noah Land Surface Model (Tewari et al.,2004). The rapid radiative transfer
model (RRTM) (Mlawer et al.,1997) and the Dudhia scheme (Dudhia,1989) for longwave and
shortwave radiative flux calculations, respectively. The Kain-Fritsch cumulus parameterization
scheme (Kain,2004) is utilized for the outer two coarse-resolution domains but is bypassed in
the finest domain (i.e., D03). The Thompson-ensemble cloud microphysics scheme is applied
for explicit cloud processes (Thompson et al.,2008; Yin et al.,2022a).

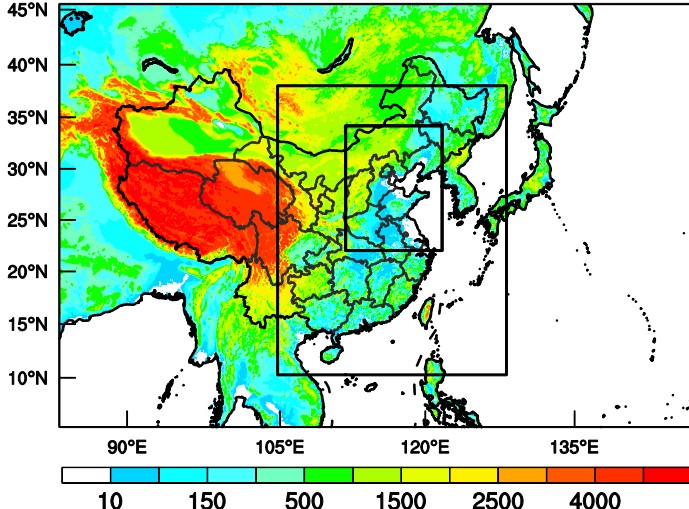

**Fig. 4** The WRF model orography (m, shadings) and the nested model domains used for
simulations with the grid sizes of 9 km (D01), 3 km (D02), and 1 km (D03).

The WRF model is integrated for 108 hours, starting from 1200 UTC 28 July 2023, with

outputs at 6-min intervals. The model outputs in the first 12 h are considered as the spin-up
process and thus are not used for the present work. The initial and outermost boundary
conditions are interpolated from the final operational global analysis of 1-degree by 1-degree
data at 6-h intervals from the Global Forecasting System of the National Centers for





Environment Prediction (NCEP). In order to force large-scale fields consistent with the driving
fields, grid analysis nudging is activated by performing the Four-Dimension Data Assimilation
(FDDA) throughout the model integration (Bowden et al., 2012; Stauffer et al., 1991). The
innermost domain (i.e., D03) outputs are validated and used for further analysis, and the
outermost domain (i.e., D01) outputs are used to demonstrate weather-scale dynamical and
thermal features. Wind profiler and surface hourly observations are provided by the National
Meteorological Information Center (NMIC) of the China Meteorological Administration (CMA)
after strict quality control.
*3.2 Model verification*

Figure 5 shows the spatial distribution of 96-h accumulated rainfall from the simulation

during the period from 0000 UTC 29 July to 0000 UTC 02 August 2023. Generally speaking,
the WRF model replicates well the spatial distribution of heavy rainfall. The heavy rainfall belt
coinciding with the orography with three rainfall cores is reproduced well, and the simulated
extreme rainfall amount compares favorably to the observed. Note that the model produces a
peak 96-h accumulated rainfall of 778 mm over the XT region, while the maximum rainfall of
1004 mm was observed over the XT region. Despite the simulation underestimates rainfall over
this region, it captures the main features of rainfall over central and North China.

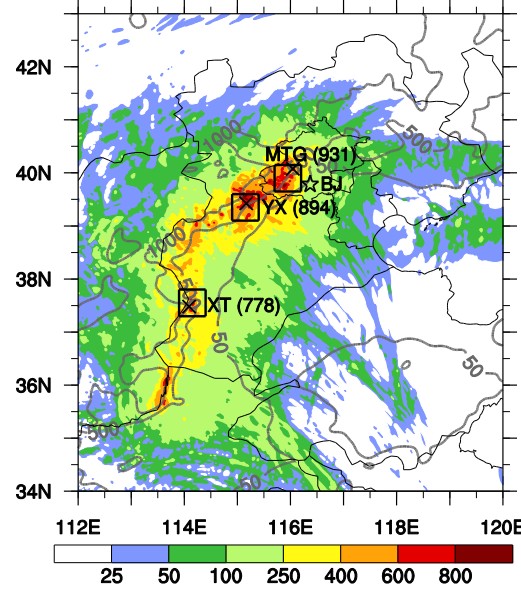


**Fig. 5** Same as in Fig. 1 but for the simulated rainfall (mm, shadings).



Figure 6 compares the spatial distribution of daily rainfalls between observations and
simulations during the period from 0000 UTC 30 July to 0000 UTC 2 August 2023. From
observations, one can see that the daily rainfalls show obvious variations. On the first day (Fig.
6a), the rainfall occurred mainly in northern Henan and southern Hebei, on the east side of the
Taihang Mountains with the rainfall cores over 250 mm. On the next day (Fig. 6b), the rainfall
extended significantly northeastward, and a new strong rainfall core occurred, covering central
Hebei Province and southwest Beijing. On the third day (Fig. 6c), rainfall was significantly
reduced in both coverage and intensity, mainly occurring in Beijing and the surrounding areas.
On the fourth day (Fig. 6d), rainfall moved eastward and weakened rapidly. It is apparent that
the model reproduces the evolutions of the rainfall, with general characteristics that are similar
to the observed (Fig. 6e-h).

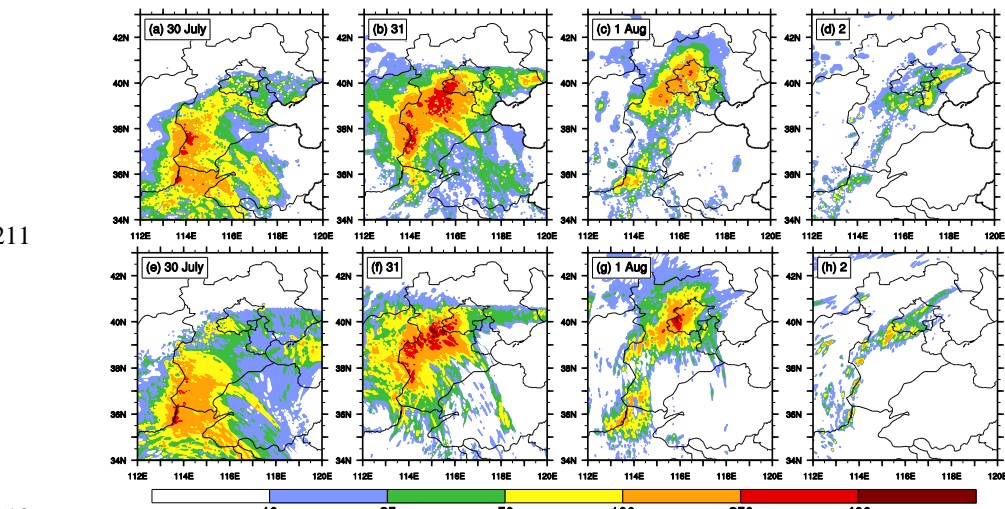


**Fig. 6** Spatial distribution of (a-d) observed and (e-h) simulated daily rainfall (mm) during the
period from 0000 UTC 30 July to 0000 UTC 2 August 2023.
Figure 7 compares the time series of hourly rainfall rates between the observed and
simulated over the MTG, YX, and XT regions. The rainfall event is characterized by long
duration, widespread coverage, and high accumulation. As has been mentioned above, the
rainfall extended from south to north, covering Henan, Hebei, and Beijing. The rainfall first
occurred in the XT region and ended near 0000 UTC 31 July 2023. As the rainfall moved
northeastward, both the MTG and YX regions occurred, ending nearly at 0000 UTC 2 August.



The observed timings of initiating and ending of the rainfall event are well replicated by the
WRF model. Besides, the observed peaks are reproduced, although there are some timing errors.
For example, the strongest rainfall occurred over the MTG region during the period from 0000
UTC to 0600 UTC 31 July. However, the simulated strongest rainfall has a 6-h lag, occurring
from nearly 0600 UTC to 1200 UTC 31 July. Overall, good agreements between the simulation
and observations are obtained in terms of the timing and location in the spatial distribution of
rainfall.

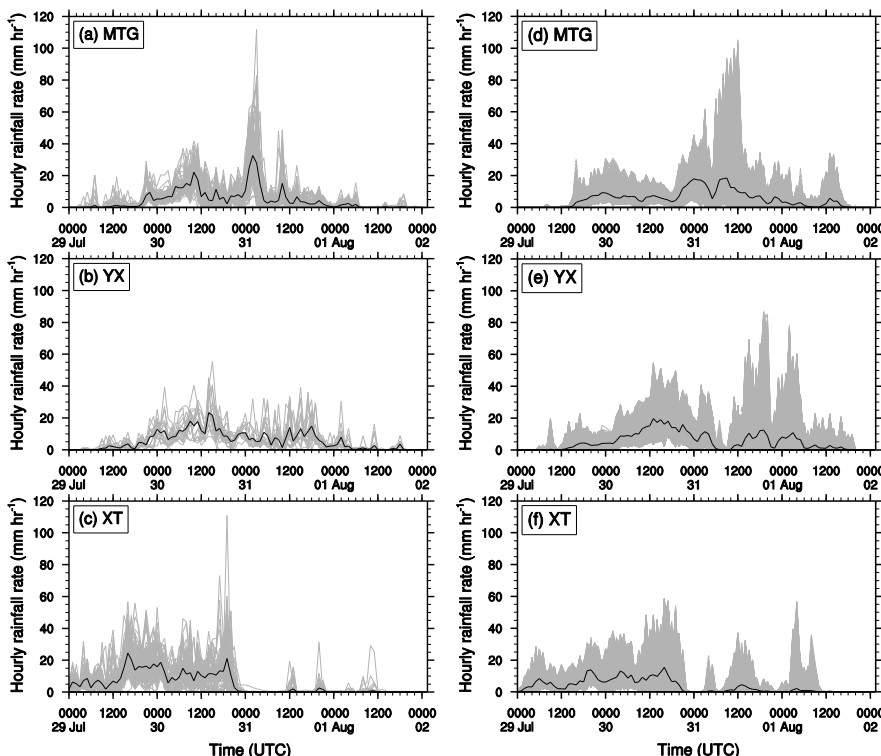


**Fig. 7** Time series of (a-c) rain gauge observations and (d-f) simulation hourly rainfall rates
(gray lines, mm hr$^{-1}$) for all stations/grid points over the (a, d) MTG, (b,e) YX, and (c,f) XT
regions during the period of 0000 UTC 29 July to 0000 UTC 02 August 2023. The black line
denotes the domain-averaged hourly rainfall rates of all stations (grid points) from observations
(simulations). (see Fig. 1 their locations).

The evolution of the simulated wind profile is presented in Fig. 8. Similar to the observed

(Fig. 2), the simulated easterly wind gradually increased from nearly 1200 UTC 29 July,
corresponding to the start of the precipitation (Fig. 6d-f). The horizontal wind experienced from
easterly to southerly except for near the ground, and then turned to southwesterly with wind
speed decreasing significantly. Overall, the variations of the simulated wind profile were
consistent with those observed, indicating that the WRF model well captured the main features
of the wind profile. Based on the wind profile and rainfall features, the simulated rainfall is
roughly divided into two stages. The shift moments (roughly marked by thick black lines) are
near at 0800 UTC 31, 2000 UTC 30, and 1600 UTC 30 July for the MTG, YX, and XT regions,
respectively.

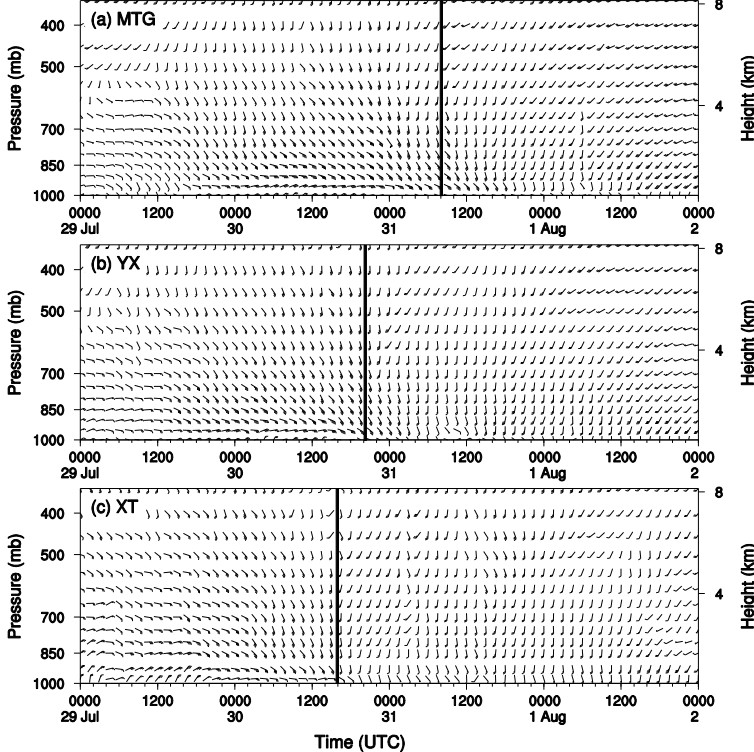

**Fig. 8** Same as Fig. 2 but for the simulated. The black lines denote wind shift from southeasterly
to southerly/southwesterly over the levels in the low to middle troposphere, which roughly
divided the rainfall into two stages.
**4. Characteristics of the rainfall event**
***4.1 Dominant dynamic processes for convection initialization***
The evolution of dynamical and thermal systems of the rainfall event in the first stage is shown
in Fig. 9. Although only a remnant vortex remained over central China at this time, typhoon
Doksuri had an important influence on the WPSH when it was strong as a super typhoon.





Several days before the rainfall event, the super typhoon Doksuri was close to the WPSH, and
the southwest WPSH edge was within the typhoon's outer region. Owing to the inflow mass
flux entering the typhoon region, and thus the southwest part of WPSH was severely destroyed
by typhoon Doksuri (Sun et al.,2015). As a result, the west boundary of the WPSH appeared to
an eastward retreat from 500 hPa to 850 hPa, showing an inclined vertical distribution on the
western boundary, especially from 700 hPa to 850 hPa. Capped by the inclined WPSH, water
vapor was mainly transported to North China through a passage nearly under 850 hPa which is
built by the typhoon remnant vortex and the tropical storm Khanun. At 500 hPa (Fig. 9a), the
WPSH (represented by the 588 isoline) covered a large part of eastern China, with an unusual
westward extension of the northwest corner to northwest China. The northwest corner extended
much further westward, compared to that before 12-h (Fig. 3). Similar patterns can be seen at
700 hPa (Fig. 7b), but the west boundary of WPSH (represented by the 316 isoline) retreated to
the East China Sea except for the northwest corner. At 850 hPa (Fig. 9c), the WPSH
(represented by the 156 isoline) completely retreated to the western Pacific, which was far away
from China.

The spatial distribution of the high PW was consistent with that of a large equivalent

potential temperature ($\theta_e$) of 344K at 500 hPa, indicating that the 334K contour covered a
relatively warm and/or wet region (Fig. 9a). Most important, the boundary of the high PW
corresponded to the large value of the potential temperature gradient over 8K on the east side
and 12K on both north and west sides. Previous studies (e.g., Rao et al.,2023) proposed that the
heavy rainfall region was closely attributed to the distributions of $\theta_e$. Although the warm and
moist conditions were favorable for precipitation, the unfavorable large-scale forcings explain
well why no deep convection was formed over this region (marked with a dashed-line box in
Fig. 9c,d). The convergence, resulting from changes in wind direction and wind speed, was
conducive to triggering convection. Consequently, the weak convergence led to weak lifting
and consequent precipitation. Since the convergence occurred at the junction of cold and warm
air masses, like a warm front rainfall, rainfalls were formed in low intensity but long duration
and widespread coverage. It is important to note that the spatial distribution of rainfall is usually
considered to be consistent with the western boundary of WPSH (i.e., the 588 isoline) at 500
hPa. However, the spatial distribution of rainfall in the present event is consistent with the dense
zone of $\theta_e$, instead of the WPSH. Therefore, in addition to the isoline 588 at 500 hPa, the spatial
distribution of $\theta_e$ needs to be given more attention in future operational forecasts.

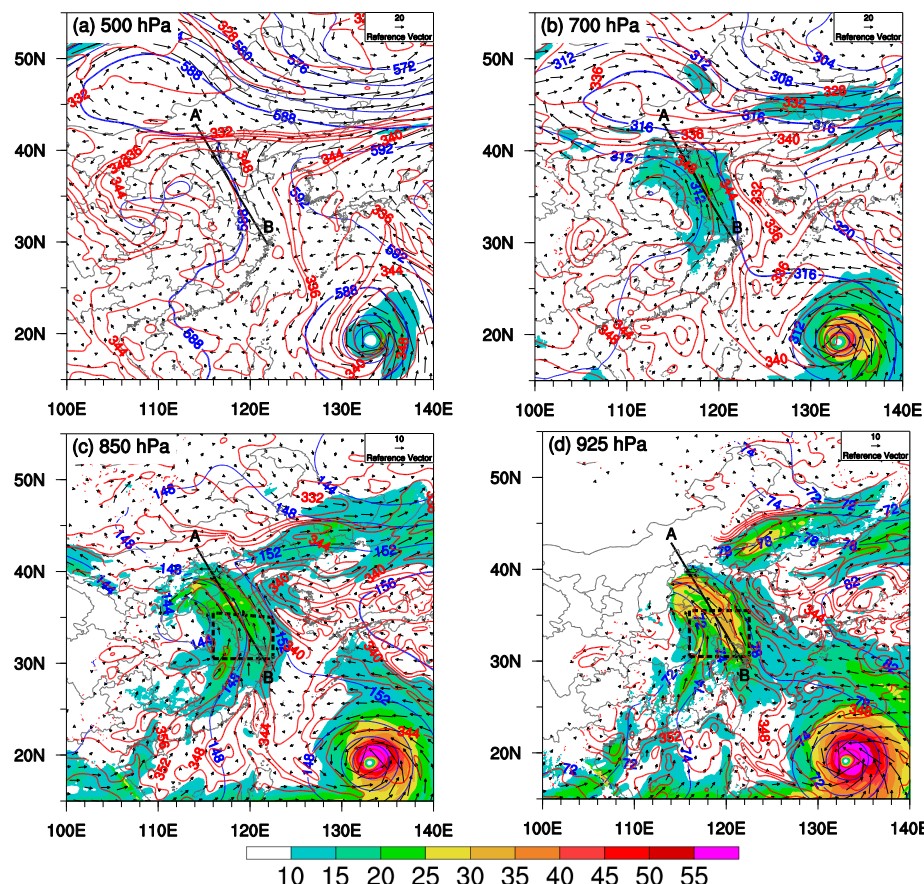

**Fig. 9** Spatial distribution of geopotential height (blue contoured at 40 gpm), equivalent
potential temperature (red contoured at 2K intervals, $\theta_e$), wind bars (a full barb is 4 m s$^{-1}$), and
water vapor flux (g s$^{-1}$ cm$^{-1}$ hPa$^{-1}$, shadings) from the model D01 at 0000 UCT 30 July 2023:
(a) 500 hPa, (b) 700 hPa, and (c) 850 hPa, and (d) 925 hPa. The isolines of 588, 316, and 156
are bolded to represent the WPSH at 500 hPa, 700 hPa, and 850 hPa, respectively. The
convergence zone of southeast and southwest flows is marked by a dashed line box in panels
(c) and (d). The thick black line A–B denotes the locations for cross-section along the water
vapor transport pathway used in Fig. 10.

The warm and moist features over North China can also be seen from the cross-section

along line A–B as shown in Fig. 10. The western orography region was controlled by cold air
mass over the levels above 3.0 km. Under the conditions, significant equivalent potential
temperature gradients were established between the warm and cold air masses, similar to a
warm front. Meanwhile, owing to the blocking of orography below 1.3 km and the strong cold



air mass above 3.0 km, only the southeasterly flows between 1.3 and 3.0 km above the sea level
can overpass the mountains. It should be noted that although the warm and moist southeasterly
flows were lifted by the orography, they could not move further upward to trigger convection
because of the local capped cold and dry air masses overhead. Consequently, convergence
mainly resulted from the changes in wind direction and wind speed led to upward motion. As
the warm and moist air was lifted, condensation occurred and thus generated precipitation. It
should be emphasized that the lifting was too weak to allow convection to be highly organized
(Fig. 10). For example, the updrafts in strong deep convective systems (e.g., Yin et al.,2020;
Yin et al.,2022c) are 5-10 times as large as the updrafts in the present event. Therefore, the weak
lifting was responsible for the rainfall in large coverage but low intensity. Besides, the
continuous and stable water vapor supply was another favorable factor for the precipitation.

Also from Fig. 10, one can see that North China was surrounded by warm dry air masses

on the east side and cold dry air masses on both north and west sides. More specifically, the air
mass at the levels above 1 km on the east side was over 3℃ warmer than surrounding regions,
but the water vapor mixing ratio ($q_v$) was less than 14 g kg$^{-1}$ (humidity was less than 70%)
because this region was controlled by the WPSH. The warm-dry cap overhead explains well
the absence of convection and rainfall over this region (cf. Figs. 5 and 6). On the north and west
sides, the air masses were dry with $q_v$ less than 2 g kg$^{-1}$. The air was over 3℃ colder than the
surrounding region except for the air near the ground. Note that warm air near the ground might
be associated with radiative heating from the ground. Capped by the cold and dry air overhead
explains why convection could not be advanced over the mountains.
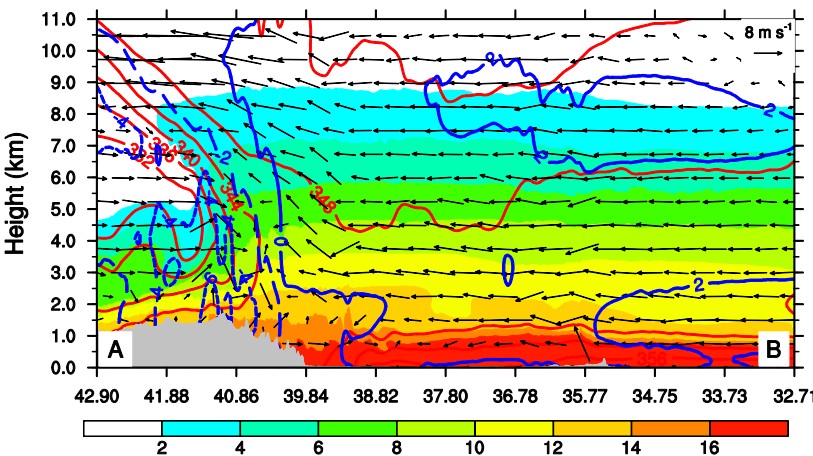


**Fig. 10** Vertical cross-section along line A–B given in Fig. 9 of temperature deviations (blue-contoured at 2°C intervals) from their level-averaged values in the cross-section, equivalent potential temperature (red-contoured at 4K intervals), water vapor mixing ratio ($q_v$, g kg$^{-1}$, shadings), and in-plane flow vectors (vertical motion amplified by a factor of 20) at 0000 UCT 30 July 2023, respectively. Gray shadings denote terrain.

In the second stage (Fig. 11), obvious differences in dynamical and thermal processes can be viewed, compared to those in the first stage (cf. Fig. 9). At 500 hPa (Fig. 11a), the WPSH further expanded westward with its western border reaching western China. It should be emphasized that the southwest part of WPSH was severely damaged by the rapid intensification of Khanun into a super typhoon. Meanwhile, as the trough deepened over northeastern China, cold air from the north poured southward. Consequently, a north-south orientated $\theta_e$ dense zone was established over eastern China. Similar patterns in $\theta_e$ and horizontal wind field can be seen at 700 hPa (Fig. 11b). However, the WPSH (represented by the 316 isoline) was further disrupted as the Khanun continued to intensify, appearing that the WPSH retreated to the East China Sea except for the northwest corner. The north-south orientated $\theta_e$ dense zone greatly prevented water vapor from transporting to North China above 850 hPa, and thus water vapor was mainly transported to North China by a shallow southeasterly flow near the ground (Fig. 11c,d). Consequently, the water vapor flux was significantly reduced (Fig. 12a). Besides, North China was dominated by southerly flows over levels above 500 hPa, and thus mid-tropospheric wind shear was significantly enhanced.


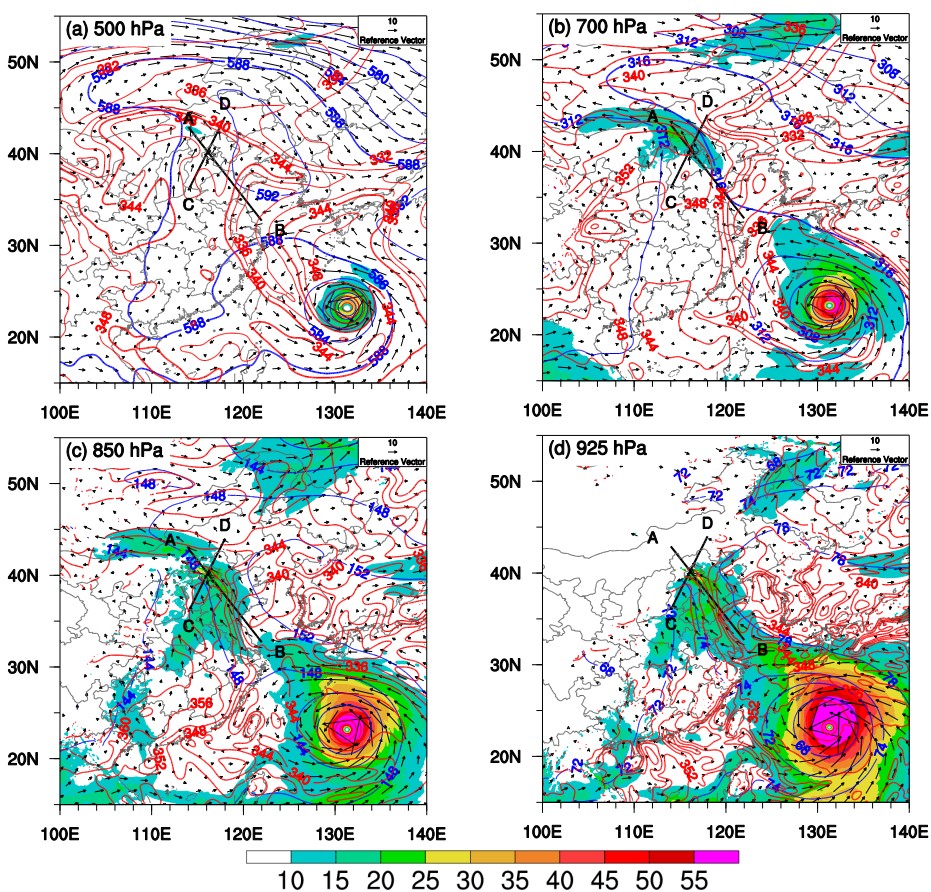

**Fig. 11** Same as Fig. 9 but for 0800 UTC 31 July 2023. Thick black lines A–B and C–D denote the locations for the cross-section in Fig. 12.

As addressed above, variations in environmental conditions caused consequent rainfall changes in nature. Especially, the shift in the wind field brought changes in thermodynamic processes and water vapor sources. Before the wind shift (Figs. 9 and 10), water vapor was mainly from the East China Sea associated with the cyclonic circulation of the typhoon remnant vortex and the tropical storm Khanun and southeasterly flow below 925 hPa. After the shift, water vapor flux was significantly reduced from both southwesterly and southeasterly flows (Fig. 11). Under the framework, convections were triggered by orographic blocking and lifting of southerly/southwesterly flows as convective instability air approached orography (Fig. 12). Unlike in the first stage, convections were further developed over mountains northward, forming deep convections (Fig. 12b). One of the reasons is that the cold air on the north side



moved northward. Comparatively speaking, the convections in the second stage are much
stronger and deeper than those in the first stage. Consequently, the rainfall intensity is increased,
compared to those in the first stage (Figs. 7d,e). The weak convections may be attributed to the
reduced water vapor supply during this period.

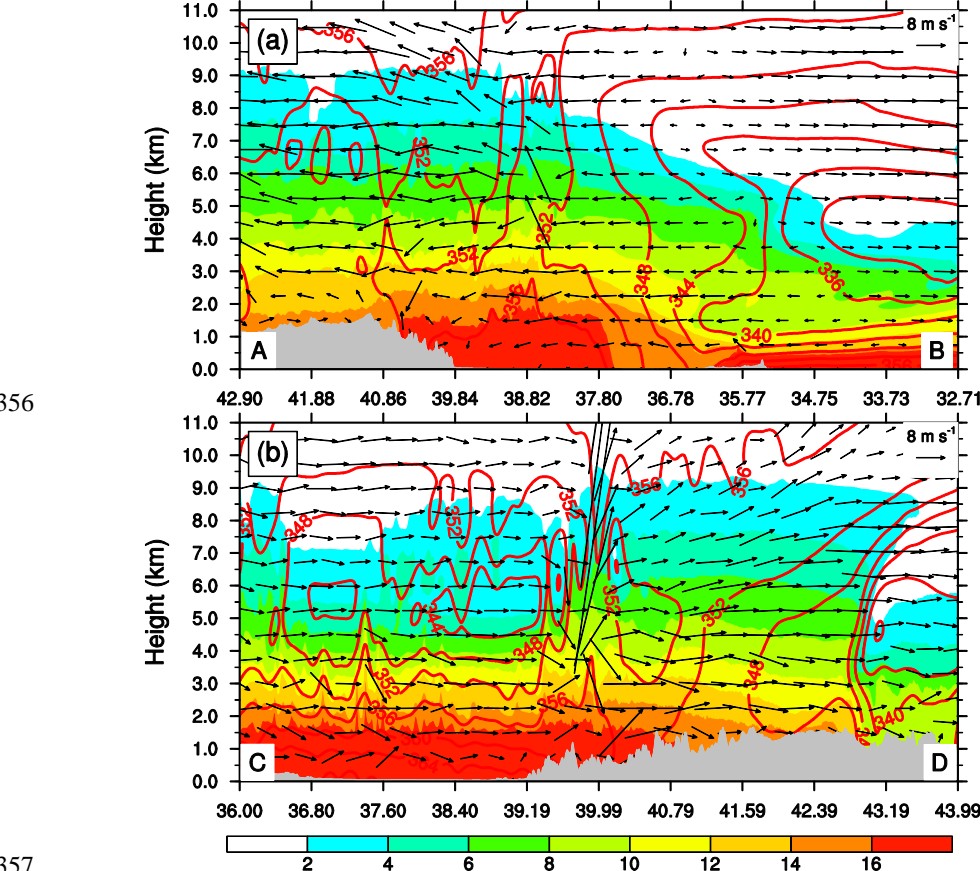


**Fig. 12** Vertical cross-sections along lines (a) A–B and (b) C–D given in Fig. 11 of equivalent
potential temperature ($\theta_e$, red-contoured at 4K intervals), water vapor mixing ratio ($q_v$, g kg$^{-1}$,
shadings), and in-plane flow vectors (vertical motion amplified by a factor of 10) at 0800 UCT
31 July 2023. Gray shadings denote terrain.
***4.2 Moisture budget***
The shift in wind direction and speed implies a change in water vapor source and rainfall
properties (Fig. 13). As stated above, water vapor was mainly from the East China Sea
associated with the cyclonic circulation of typhoon Khanun before the wind shift, and was



fueled by the southeasterly flow below 925 hPa. After the shift, the water vapor supply was
significantly reduced from both southwesterly and southeasterly flows. Figure 13 shows the
time-height cross-sections of moisture flux across eastern, southern, western, and northern
boundaries and total lateral boundary moisture flux for the MTG region. The moisture flux is
calculated as
$$QFlux = \int_0^L q_v \vec{v} \, dl \,.$$

Here, $QFlux$ is moisture flux across one of the four boundaries, and $q_v$, $\vec{v}$, and $L$ are water
vapor mixing ratio, wind vector, and the length of the boundary, respectively. The TOT is a
summation of the $QFluxs$ from the four boundaries by taking inward(outward) as
positive(negative).

One can see that the MTG region experienced vigorous lower-to-middle level inward

(outward) moisture fluxes across their eastern and southern (western and northern) boundaries.
For the eastern boundary (Fig. 13a), the inward moisture flux began to increase gradually from
0000 UTC 29 July, with the maximum values over 13,500 kg kg$^{-1}$ m$^2$ s$^{-1}$ occurring between
1200 UTC 30 and 0000 UTC 31 July 2023. Then, the inward flux moisture decreased rapidly
and even transformed to the outward flux at 0000 UTC 1 August 2023. The inward moisture
flux was mainly concentrated below 3 km above the sea level because upper levels were capped
by the warm dry air masses associated with the WPSH (cf. Figs. 9 and 10). However, owing to
weak lifting over, most of the water vapor flowed out through the western boundary (Fig. 13b).
Meanwhile, part water vapor was transported in this region from the southern boundary except
for the lower levels during 0000 UTC 30 to 0000 UTC 31 July 2023 (Fig. 13d). The outward
flow water vapor resulted from the northeasterly around flow due to the blocking of the Yanshan
Mountains. Similar patterns can be seen in the northern boundary with almost the same outward
water vapor flux (Fig. 13c). The temporal evolution of the water vapor flux across the eastern
boundary is consistent with that of rainfall over this region (Figs. 13a and 7d), suggesting that
rainfall formation was dominated by the inward of water vapor from the eastern boundary.
Overall, the inward net moisture fluxes were concentrated in the lower troposphere between 0.5
km and 1.5 km (Fig. 13e), suggesting that most of the water vapor was consumed at this layer
by condensation. Despite the high water vapor flux, the water vapor-rich layer is too thin (nearly
1 km) to be favorable for the formation of heavy rainfall. Similar patterns can be found over
both YX and XT regions (not shown), although there were temporal and quantitative differences.

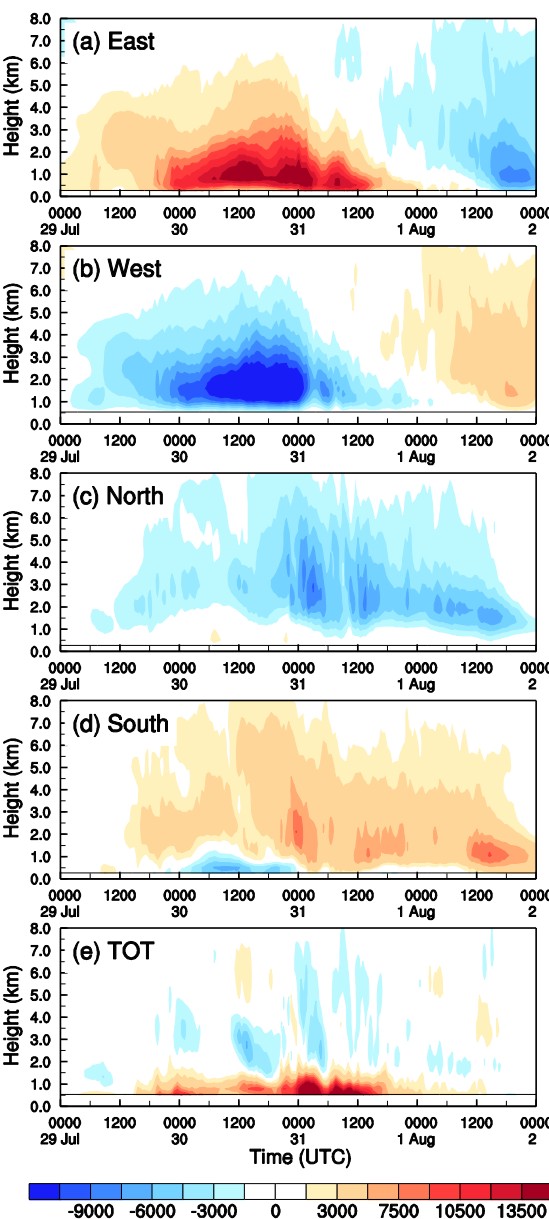


**Fig. 13** Time-height cross sections of moisture fluxes (kg kg$^{-1}$ m$^2$ s$^{-1}$) through the (a) eastern,

(b) western, (c) northern, and (d) southern boundaries of the MTG region in Fig. 1; (e) TOT
provides the total net moisture flux of all boundaries.





In the second stage, the north-south orientated $\theta_e$ dense zone greatly prevented water vapor
from being transported to North China by southeasterly flows from the East China Sea, and
thus water vapor was mainly transported to North China across the south boundary (Figs. 13b,d).
Unlikely, the water vapor was mainly provided by southeasterly(southwesterly) flow
below(above) 500 hPa. Note that the water vapor flux amount was significantly reduced (Fig.
13). Despite the thickening of the water vapor flux layer associated with the
southerly/southwesterly flows, the water vapor flux is much less, compared to the first stage.
Therefore, the wind shift had strong effects on the reduction in water vapor flux and consequent
rainfall over North China. The same results can also be obtained in the YX and XT regions (not
shown). It is worth emphasizing that strong hourly rainfalls occurred during the wind shift
period (cf. Figs. 2, 7, and 8), suggesting that the changes in wind direction enhanced wind shear
and thus promoted the development of convections and consequent precipitation under moisture
and instability conditions (Chen et al.,2015; Rotunno et al.,1988; Schumacher and Rasmussen,
2020). Therefore, it is important to pay special attention to environmental wind alterations in
future remote rainfall forecasts.
***4.3 Properties of convection***
Figure 14 shows the temporal evolution of maximum upward motion and radar reflectivity
over the MTG region during the rainfall period from 0000 UTC 29 to 0000 UTC 2 August 2023.
In the first stage (i.e., before 0800 31 July), most of the maximum updrafts were almost less
than 3 m s$^{-1}$. Owing to the weak updrafts, the storm did not stretch as high as typical convective
systems over North China, with hydrometeors concentrated on the levels with temperature
above 0℃ (Fig. 14a). As addressed above (Fig. 10), weak updrafts were attributed by the
unfavorable large-scale forcings. The vertical distribution of hydrometeor indicates that the
warm rain processes were dominant in the persistent rainfall event. The result is consistent with
the water vapor consumed layer between 0.5 km and 1.5 km (Fig. 13e). Unlikely, the maximum
updraft was over 11 m s$^{-1}$ in the second stage (i.e., after 0800 31 July), which is much stronger
than that in the first stage (Fig. 14). Correspondingly, the radar reflectivity penetrated through
the 0℃ level with a cloud top exceeding 12 km, indicating that both warm and cold rain
processes were active in this stage. Correspondingly, the intensity of hourly rainfall increased
significantly, with the maximum value exceeding 100 mm (Fig. 7d). Comparatively speaking,
there are larger strong convective areas in the second than those in the first stage. The same
features were also found in the regions of YX and XT (not shown). Unlike the usual short-
duration heavy rainfall in North China (Mao et al.,2018; Xia and Zhang,2019; Yin et al.,2022b),
this precipitation was mainly dominated by warm cloud processes (Fig. 14). As addressed above,
the weak updrafts but warm-moist air were responsible for persistent rainfall but low intensity.
A detailed analysis of cloud microphysical processes for this event will be given in a
forthcoming study, in which all microphysical source and sink terms will be explained.

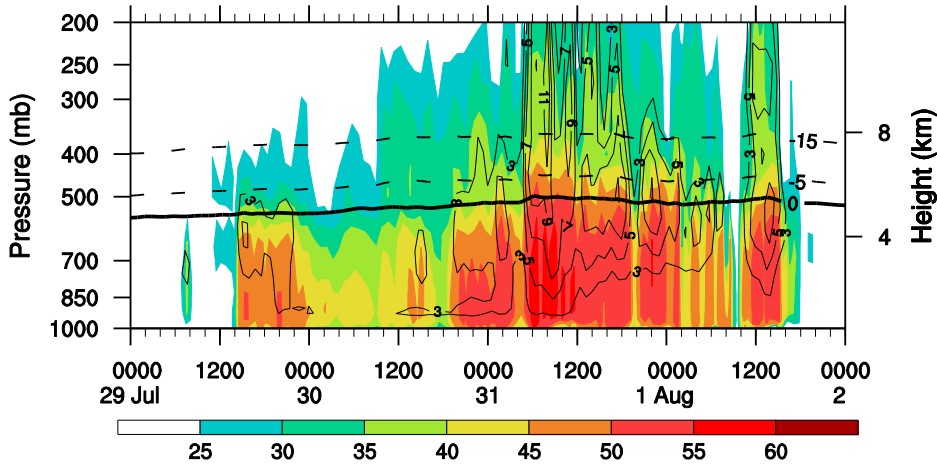


**Fig. 14** Time-height cross-section of domain maximum radar reflectivity (dBZ, shadings) and
upward motion (contoured at 2 m s$^{-1}$) taken from MTG region during the period from 0000
UTC 29 to 0000 UTC 2 August 2023. The isothermal lines denote the 0°C (the melting layer),
-5°C, and -15°C levels, respectively.
**5. Conclusions and outlook**
In this study, we examined the convective initiation and subsequent persistent heavy
rainfall over North China during the period from 29 July to 2 August 2023 with observations
and simulations with the WRF model. From observations, the rainfall was featured by long
duration and widespread coverage but low intensity, like a warm front rainfall. Firstly, the
persistent heavy rainfall event was reproduced by the WRF model. Further analysis based on
the simulations shows that this persistent precipitation was caused by a combination of a
remnant vortex originating from typhoon Doksuri(2305), the tropical storm Khanun(2306), the




west Pacific subtropical high (WPSH) with an unusual westward extension of the northwestern
corner, and stable cold dry air from over northern China.
According to the simulated wind profiles and rainfall features, the persistent heavy rainfall
event was divided into two stages. Figure 15 summarizes the synoptic-scale forcings and
possible dynamic mechanisms for the persistent heavy rainfall. In the first stage (Fig. 15a), a
water vapor transportation passage was built by a typhoon remnant vortex and a tropical storm
Khanun, providing a stable warm moist water vapor supply. Several days before the rainfall
event, the southwestern WPSH was within the typhoon Doksuri's outer region, and thus the
southwestern WPSH was destroyed by the tropical storm Doksuri. It appears that the west
boundary of the western Pacific subtropical high (WPSH) retreated eastward from 500 hPa to
850 hPa, showing an inclined vertical distribution on the western boundary, especially from 700
hPa to 850 hPa. Capped by the inclined WPSH, water vapor was mainly transported to North
China through a water vapor passage under nearly 850 hPa (Fig. 10). Although the warm and
moist regions were favorable for precipitation over North China, organized strong convective
systems were seldom because of the absence of unfavorable large-scale conditions. At the same
time, the orography in the west of North China was controlled by dry cold air mass over levels
above 3.0 km. Owing to the blockings of orography below 1.3 km and the strong cold air mass
above 3.0 km, only the southeasterly flows between 1.3 and 3.0 km above the sea level can
overpass the mountains. Although the warm and moist southeasterly flows were lifted by the
orography, they could not go further upward to trigger convections because of the locally
capped cold and dry air masses overhead. Under the conditions, significant equivalent potential
temperature gradients were established between the warm and cold air masses, similar to a
warm front. Consequently, convergence mainly resulted from the changes in wind direction and
wind speed led to upward motion. As the warm and moist air was lifted, condensation occurred
and thus generated precipitation. However, the lifting was too weak to allow convection to be
highly organized (Fig. 14), leading to the rainfall in low intensity but large coverage. Besides,
the continuous and stable transportation of water vapor provided by tropical storm Khanun
ensured stable precipitation over a long period of over 80 h. Therefore, this event shows similar
rainfall features to those of a warm front rainfall with a long duration and widespread coverage





but low intensity.

In the second stage (Fig. 15b), the WPSH further expanded westward at 500 hPa, with its

western border reaching western China. However, the southwest part of WPSH was further
damaged by the rapid intensification of Khanun into a super typhoon. Consequently, the
embedded warm-dry cover associated with the tilted WPSH was significantly thinned, favoring
convective development. Meanwhile, as the trough deepened over northeastern China, cold air
from the north poured southward. Consequently, a north-south-orientated equivalent potential
temperature ($\theta_e$) dense zone was established over eastern China, which greatly prevented water
vapor from being transported to North China (Fig. 12a). However, owing to the clockwise
rotated southeasterly flow, a deep southerly (southwesterly) flow was built over North China.
Convections were triggered by orographic blocking and lifting of southerly/southwesterly flows
as convective instability air approached orography. Unlike the first stage, the convections were
further developed over mountains northward, forming deep convections. It should be noted that
the northward-moved cold air on the north side was another favorable condition. Therefore, the
convections in the second stage are much stronger and deeper than those in the first stage,
although water vapor flux is smaller than in the second period. Consequently, the rainfall
intensity is increased, compared to that in the first stage. Correspondingly, both warm and cold
rain processes were active in the second stage, while warm rain processes were dominant in the
first stage.



**Fig. 15** (a) Three-dimensional diagram of the mechanisms for the persistent heavy in the first
stage. Several distinct synoptic systems, including the tropical storm Khanun(2306), a
remnant vortex originating from the typhoon Doksuri(2305), quasi-stationary cold dry air
masses, and an abnormal western Pacific subtropical high (WPSH) with inclined vertical
distribution on the western boundary (thick black line). Blue lines marked with 588 and 156
represent the WPSH at 500 hPa and 850 hPa, respectively. Red lines denote the spatial
distribution of equivalent potential temperature ($\theta_e$) dense zone between 336 K and 344K. At
850 hPa, black arrows indicate jets with wind speed over 12 m s$^{-1}$, and shadings denote water
vapor flux. Orange shadings imply 96-h accumulated rainfall over 200 mm; blue contours
denote sea level pressure; gray arrows denote surface (i.e., $z$ =10 m) horizontal wind with
wind speed over 5 m s$^{-1}$, and black contours indicate orography (m). (b) Same as (a) but for
rainfall in the second stage.

In this study, we have gained principal results of the persistent heavy rainfall event. It is

important to note that the spatial distribution of rainfall is usually considered to be consistent
with the western boundary of WPSH (i.e., the 588 isoline) at 500 hPa. In the present event, the
spatial distribution of rainfall is consistent with the dense zone of $\theta_e$, rather than the western
boundary of WPSH. Therefore, in addition to the 588 isoline, the spatial distribution of $\theta_e$ needs
to be given more attention in future operational forecasts. Besides, we should give weight to
environmental wind shifts, which may lead to changes in convections and the nature of
consequent precipitation. Although reasonable dynamic mechanisms for the present persistent
heavy rainfall have been proposed, there are still several questions that need to be answered.
Among those, more work is required to understand detailed cloud and precipitation processes.
In addition, diagnostic and budget analyses will be conducted to understand how the orography
facilitates the generation of the rainfall belt with three rainfall cores along the mountains.
Nevertheless, the concept of synoptic-forcing-based forecasting is discussed as it might apply
to a broader spectrum of forecast events than just over North China.

**Code and data availability**
The source code of the Weather Research and Forecasting model (WRF v4.1.3) is available at
https://github.com/wrf-model/WRF/releases (last access 1 August 2024). The National Centers



for Environmental Prediction (NCEP) Global Forecast System one-degree final analysis data at
6 h intervals used for the initial and boundary conditions for the specific analyzed period can
be downloaded at https://rda.ucar.edu/datasets/d083002/ (last access 1 August 2024). Modified
WRF model codes and all the data used in this study are available from the authors upon request.

**Author contributions**
Conceptualization: JY, JS, and XL; methodology: JY and JS; data curation: JY and FL; writing
– original draft preparation: JY, and FL; writing – review and editing: JY, ML, RX, XB, and JS;
project administration: XL; funding acquisition: JY and XL. All authors have read and agreed
to the published version of the paper.

**Competing interests**
The contact author has declared that none of the authors has any competing interests.

**Acknowledgments**
The authors acknowledge the use of the NCAR Command Language (NCL) in the preparation
of figures.

**Financial support**
This study is jointly supported by the National Key R&D Program of China
(2022YFC3003903), National Natural Science Foundation of China (42075083), Open Project
Fund of China Meteorological Administration Basin Heavy Rainfall Key Laboratory
(2023BHR-Z03), and the Development Foundation of Chinese Academy of Meteorological
Sciences (2019KJ026).



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
