# Peer review of "The miscellaneous synoptic forcings in the four-day widespread extreme rainfall event over North China in July 2023"

_Natural Hazards and Earth System Sciences, 2024_

## Author Comment (AC1)

**Response to Reviewer 2's comments**

*[General] In this study, authors examined the convective initiation and subsequent persistent heavy rainfall over North China during the period from 29 July to 2 August 2023 with station observation data and WRF model simulations. From observations, it is found that the rainfall was featured by long duration and widespread coverage but low intensity, like a warm front rainfall. Further analyses based on the WRF model simulations show that this persistent precipitation was caused by a combination of a remnant vortex originating from typhoon Doksuri(2305), the tropical storm Khanun(2306), the west Pacific subtropical high (WPSH) with an unusual westward extension of the northwestern corner, and stable cold dry air from over northern China. These results are important for understanding the reasons of this extreme precipitation event occurring over North China. But there are some flaws in the manuscript which are needed to improve. The comments are as follows:*

**Response:** Thank you very much for agreeing with us on the intention of this manuscript. We appreciate you for providing valuable comments and constructive remarks, which have helped improve our manuscript significantly.

*1.In the title of the manuscript, "miscellaneous synoptic forcings" is not reasonable. Actually authors only analyzed the remnant vortex originating from typhoon Doksuri(2305), the tropical storm Khanun(2306), the west Pacific subtropical high (WPSH) with an unusual westward extension of the northwestern corner, and stable cold dry air from over northern China. These factors are only the synoptic circulation patterns, not "forcings";*

**Response:** Thank you very much for your kind comments. In this study, we investigated the roles of different weather systems within the atmosphere in this extreme precipitation event. Indeed, as you pointed out, they are not "forcings" and, the title has been therefore revised to "**The unique features in the four-day widespread extreme rainfall event over North China in July 2023**"

*2.Line 67:"such large-scale weather conditions", what is such large-scale weather conditions? It is not clear;*

**Response:** Thanks for the kind reminder. We revised the sentence as follows:

"Previous studies (e.g., Hirata and Kawamura,2014; Gao et al.,2022; Yang et al., 2017) pointed out that large amounts of water vapor brought by a typhoon over the

North Pacific were favorable for heavy rainfall generation in eastern China."

*3.Line 74:"surface rainfall",surface should be deleted;*

**Response:** Thanks. It has been deleted.

*4.Line 85:"emerged in this precipitation",probably there is something wrong in this sentence;*

**Response:** To make it clearer, we revised this sentence as follows.

"Although operational forecasts gave reasonable results at that time, several unusual features were found to exist in this extreme rainfall event."

*5.Line 101-103:"The spatial distribution of heavy rainfalls is consistent with the orography of the Yanshan Mountains on the north and the Taihang Mountains on the south, suggesting that the heavy rainfall may be associated with the orography." Generally speaking, the spatial distribution of heavy rainfalls is consistent with the orography, but for this event, there are only three heavy rainfall centers near MTG、 YX and XT, they are not distributed with Yanshan and Taihang Mountains;*

**Response:** Thank you very much for pointing this out. The spatial distribution of the rain belt with three heavy rainfall cores is consistent with the orography of the

Yanshan Mountains on the north and the Taihang Mountains on the south. We revised the sentence as follows.

"The spatial distribution of rain belt with three heavy rainfall cores is consistent with the orography of the Yanshan Mountains on the north and the Taihang Mountains on the south, suggesting that orography plays an important role in the precipitation."

*6.Is Xiangtai (XT) right? It seems Xingtai(XT);*

**Response:** It is corrected. Thank you!

*7.How to identify the wind direction in Figure 2?*

**Response:** Usually, the wind variations within the planetary boundary layer have an important effect on precipitation. Therefore, we pay attention to the wind field in levels below 2 km.

*8.Line 139-142:"One can see that the large-scale flow patterns exhibited a coexistence of a remnant vortex originating from typhoon Doksuri(2305) and*

*tropical storm Khanun(2306). The former weakened significantly into a vortex at*
*this time, while the latter was in the rapid development stage." It is known from this*
*sentence that tropical storm Khanun(2306) is in the rapid development stage, so the*
*circulation associated with tropical storm Khanun(2306) is not remnant vortex.*

**Response:** Sorry for the misunderstanding. This part has been revised as follows:

"One can see that the large-scale flow patterns exhibited a coexistence of the tropical storm Khanun(2306) with a remnant vortex originating from typhoon

Doksuri(2305). Note that the Khanun was in the rapid development stage, while the vortex weakened significantly at this time."

*9.In the caption of Fig. 6, (a-d) observed and (e-h) simulated daily rainfall are not*
*consistent with that in the text;*

**Response:** Thanks for pointing this out. The text has been updated. We went through the entire manuscript to eliminate such mistakes.

*10.The caption of Figure 7: How many stations/grid points over the (a, d) MTG, (b,e)*
*YX, and (c,f) XT regions used to draw these figures?*

**Response:** Thank you very much for the kind suggestion. The stations/grid points are provided in the captions.

"In total, 74, 19, and 67 observations are used for (a) MTG, (b) YX, and (c) XT,
respectively. For the simulation, there are (d) 2296, (e) 2365, and (f) 2420 grid
points."

*11.Line 240-241:Based on the wind profile and rainfall features, the simulated*
*rainfall is roughly divided into two stages?What is the rationale to divide the*
*precipitation into two stages? For this event, the rainfall belt moved from south to*
*north with the Typhoon Doksuri movement, so it can not be divided into two stages;*

**Response:** Thanks for your comments. Yes, as pointed by you, the rain belt moved from south to north during the four days. However, except for the remnant vortex originating from typhoon Doksuri(2305), the rainfall was also influenced by the tropical storm Khanun(2306). In the early stage (see Fig. 9 in the manuscript), the remnant vortex was active and the tropical storm Khanun was far away from China. As a result, water vapor is mainly provided by the counterclockwise southwesterly flow with the vortex. In the late stage (Fig. 12), the vortex weakened significantly, and the typhoon Khanun developed rapidly and approached China. Water vapor was mainly supplied by the southeasterly flow associated with typhoon Khanun. Therefore, the rainfall was roughly divided into two stages according to wind profiles and rainfall features.

*12.Line 353-354*:"*Consequently, the rainfall intensity is increased, compared to*

*those in the first stage (Figs. 7d,e). The weak convections may be attributed*

*to ……*",*rainfall intensity increase is inconsistent with the weak convections;*

**Response:** Thanks for pointing this out. This sentence is too abrupt and so has been removed from the revised version.

*13.The sub-title of Part 4 "Characteristics of the rainfall event" is not reasonable.*

*The contents of this part are only physical quantity diagnoses, not related to the*

*miscellaneous synoptic forcings.*

**Response:** Thanks for this point. The sub-title has been changed into "Unique features for the extreme rainfall".

We appreciate you very much for your positive and constructive comments and
suggestions on our manuscript, which are valuable in improving the quality of our
manuscript.

---

## Author Comment (AC2)

**Response to Reviewer 1's comments**

*[General Comments] In the present work, the persistent heavy rainfall event during the period from 29 July to 2 August 2023 ("23·7" event) was examined, focusing on synoptic forcings. Special attention was paid to an inclined vertical distribution on the western boundary was figured out, except for the remnant vortex originating from typhoon Doksuri(2305) and tropical storm Khanun(2306). Although previous studies considered the influence of the western Pacific subtropical high (WPSH) on heavy rainfall over north China, little has been documented on a tilted western boundary of WPSH. Interestingly, warm-dry cap overhead associated with the WPSH explains well the absence of convection and rainfall along the western boundary of WPSH. Overall, this study broadens the horizon of extreme precipitation in North China.*

*The study is well-designed and clearly organized, making it both interesting and relevant. Prediction of extreme rainfalls is an important field with significant implications for public safety. I would like to recommend publishing this paper in NHESS after addressing the relatively minor comments and clarifications below.*

**Response:** Thank you very much for your kind words and positive comments on our manuscript. In the following sections, you will find our responses to each of your points and suggestions. We are grateful for the time and energy.

Specific comments:

(1) Although wind bars are given in Fig. 2, it is still hard to get low-level jet(s). It is recommended to draw the contour lines of wind speed or color the wind bars with wind speed for low-level jets.

**Response:** This is an excellent suggestion. We have redarw Fig. 2, and updated the text accordingly. For your convenience, the updated figure is shown as follows.

[Figure]

**Fig. R1.** Temporal evolution of wind profile (a full barb is 4 m s⁻¹, and shadings denote wind speed over 10 m s⁻¹) from observations near (a) MTG, (b) YX, and (c) XT during the period of 0000 UTC 29 July to 0000 UTC 2 August 2023. Note only the wind profile below 5 km above the ground can be observed due to the limitation of the instrumentation near Xingtai (XT).

*(2) Lines 240-243: "The shift moments (roughly marked by thick black lines) are near at 0800 UTC 31, 2000 UTC 30, and 1600 UTC 30 July for the MTG, YX, and XT regions, respectively". Why wind shift occurred at different moments? It is necessary to indicate the difference in the weather system during these two periods.*

**Response:** In the present event, the wind field was significantly influenced by Typhoon Khanun (2306). As the typhoon gradually moved northwestward and the vortex weakened, the first to be affected was Xingtai (XT) in the south, then Yixian (YX) in the centre and finally Mentougou (MTG) in the north. Therefore, the wind shift occurred at different moments.

*(3) Line 434: "this precipitation was mainly dominated by warm cloud processes*

*(Fig. 14)." The result is derived from the simulation. Can similar structural features*

*be observed from radar observations?*

**Response:** Yes. The temporal evolution of the observed radar reflectivities over

Mentougou (MTG) is shown in Fig. R2. It can be seen that strong reflectivities are mainly in the lower and middle troposphere, indicating active warm cloud processes.

[Figure]

**Fig. R2.** Time series of radar observations (top) and rain gauge rainfall rates (bottom, mm/6min) near Mentougou (MTG).

*(4) Since terrain plays an important role in the precipitation, it is recommended to*

*take terrain over 1000 m into account in the three-dimensional diagram (i.e., Fig.*

*15).*

**Response:** Thank you for this great suggestion. It is hard to insert a three-dimensional terrain in the diagram. Therefore, the topography over 1000 m is superimposed on the map, as shown by the black contours in Fig. 15.

*(5) Technical comments: Please consider adjusting your reference list with the*

*manuscript preparation guidelines of NHESS.*

**Response:** Thanks for the kind suggestions. The references have been updated according to the NHESS.

We appreciate you very much for your positive and constructive comments and suggestions on our manuscript, which are valuable in improving the quality of our manuscript.

---

## Author Response (AR2)

**Response**

Dear Dr. Gregor Leckebusch,

We are deeply grateful to you and the reviewers for the valuable comments and suggestions, which helped us improve the manuscript greatly. Below, we explained how the comments and suggestions are addressed point-by-point, and make notes of the revisions we have made in the updated manuscript.

Thanks very much for your handling of our submission again, and we are looking forward to hearing from you soon.

Sincerely yours,

Dr. Jinfang Yin

yinjf@cma.gov.cn

State Key Laboratory of Severe Weather, Chinese Academy of Meteorological Sciences, Beijing 100081, China

23 December 2024

**Response to editor (Gregor C. Leckebusch)**

*[General Comments]* I recommend minor revisions and suggest you to add a specific section to the manuscript explaining in detail how the manuscript addresses one core aim of our journal: to the study of the evolution of natural systems towards extreme conditions, and the detection and monitoring of precursors of the evolution. Especially, I would recommend you clarify how your submission separates from a localised case studies with no broader implications. The latter separation being a necessary condition for a publication in NHESS.

**Response:** Thank you very much for your kind suggestion. We apologize for missing your suggestion in the previous revision. In response to the comment we have added a detailed description of the core aim of the NHESS journal and special words are added to explain why we focused on a widespread extreme rainfall event, rather than a localized case in the *Introduction* section.

During the period from 29 July to 1 August 2023, North China experienced devastating rainfall. Despite the rainfall in low intensity, it was long-lasting and widespread, resulting in large accumulated rainfall. Overall, the average accumulated rainfall over North China (including Beijing, Tianjin, and Hebei province) was 175 mm, which was approximately 1/3 of the average annual precipitation in this region. There were 3 rainfall cores over 700 mm, with the maximum one over 1000 mm. Flooding from this event affected 1.3 million people, bringing severe human casualties and economic losses. The sustained heavy rainfall over Beijing left 33 people dead and 18 missing persons. Compared to a localized case, this rainfall event was characterized by stronger accumulated rainfall, longer duration, and more severe human casualties and economic losses. Although operational forecasts gave a reasonable spatial distribution of precipitation at that time, the precipitation intensity was underestimated significantly. Therefore, the topic, aiming at the evolution of natural systems towards extreme rainfall, is in the scope of NHESS.

**Response to Referee #1**

Thank you very much for your previous comments that helped us improve this manuscript.

---

## Author Response (AR3)

**Response**

Dear Dr. Gregor Leckebusch,

We are deeply grateful to you for your valuable comments and suggestions, which helped us improve the manuscript greatly. We do invite a colleague who is proficient in English to revise the English writing of the manuscript to polish our article. We hope the revised manuscript could be acceptable.

Thank you very much, and we are looking forward to hearing from you soon.

Sincerely yours,

Dr. Jinfang Yin

yinjf@cma.gov.cn

State Key Laboratory of Severe Weather, Chinese Academy of Meteorological Sciences, Beijing 100081, China

23 January 2025

---

## Author Response (AR4)

**Response**

Dear Dr. Gregor Leckebusch,

We appreciate you for your precious time in reviewing our paper and providing valuable comments, which helped us improve the manuscript greatly. The authors have carefully considered the comments and tried our best to address every one of them. We hope the manuscript after careful revisions meets your high standards, and the revised manuscript could be acceptable. Below we provide the point-by-point responses.

Thank you very much, and we are looking forward to hearing from you soon.

Sincerely yours,

Dr. Jinfang Yin

yinjf@cma.gov.cn

State Key Laboratory of Disaster Weather Science and Technology, Chinese Academy of Meteorological Sciences, Beijing 100081, China

20 February 2025

a) "the formation of extreme rainfalls over this region" ==> rainfall events?

**Response**: Yes. It has been revised according to your suggestion. Thank you!

b) Please clarify the numbers behind the tropical cyclone/storm name. I guess this is the CMA ID number? Please give the full original reference.

**Response**: Thanks for your kind reminder. The tropical cyclone names are provided by a special committee under the World Meteorological Organization (WMO). As we see that the typhoon names are repeated year by year, they are distinguished by a corresponding number to the year followed by the occurrence order within this year denoted by the corresponding ordinal numeral behind the tropical cyclone/storm name. Taking the typhoon Doksuri (2305) as an example, it indicates the fifth typhoon Doksuri occurring in 2023. Using the name and number, we can easily find out the information about this typhoon.

c) "western Pacific subtropical high (WPSH)" ==> Western North Pacific Subtropical High (WNPSH)?

**Response**: Yes. It has been revised according to your suggestion. Thank you.

d) "life history" ==> life cycle?

**Response**: Yes. It has been revised according to your suggestion.

e) "the western boundary of the western Pacific subtropical high (WPSH) was destroyed" meaning unclear

**Response**: In the updated manuscript, it has been modified to "the western part of the Western North Pacific Subtropical High (WNPSH) weakened". Thank you very much.

f) "an inclined vertical distribution" of what?

**Response**: It has been modified therein to "The marginal zone of this subtropical high became then inclined below 500 hPa". See our updated manuscript for further details.

g) "Therefore, convections were limited" ==> Therefore, convection was limited

**Response**: Yes. Thank you! It has been revised accordingly.

h) "the orography in the west of North China was controlled by cold air"; unclear: orography cannot be controlled by air?

**Response**: Thank you very much for your nice reminder. It has been changed to "The mountain land in the western part of North China was occupied by cold air masses" in the updated manuscript.

i) "shallow southeasterly layer" meaning unclear: a layer southeast of what…?

**Response**: Thank you very much for your nice reminder. It has been changed to "only a thin layer of southeasterly wind (between 1.3 and 3.0 km) in the updated manuscript.

j) "Under this framework" meaning unclear

**Response**: It has been changed to "Under this regime" in the updated manuscript. Thank you very much.

k) "WPSH was further destroyed" meaning unclear

**Response**: Thank you very much for your nice reminder. It has been changed to "WNPSH was further weakened" in the updated manuscript.

l) "Consequently, convections triggered" ==> convection triggered …

**Response**: Thanks for pointing this out. Revised accordingly.

m) "by orographic blocking can move upward": meaning unclear: extend upward?

**Response**: Thank you very much for your nice reminder. Yes, you are right. It has been changed to "The convections triggered by orographic blocking are able to extend upward" in the updated manuscript.

n) "Comparatively speaking"; meaning unclear

**Response**: Thank you very much for your nice reminder. It has been changed to "Generally speaking"

o) "The results gained herein may shed new light on better understanding and forecasting of long-lasting extreme rainfall." Meaningless: what was gained?

**Response**: It has been deleted in the updated manuscript.

---

## Author Response (AR5)

**Response**

Dear Editor,

    Thank you very much for your prompt processing and editing of our manuscript. All required files have been uploaded. In case any questions arise, Please keep me informed.

Sincerely yours,

Dr. Jinfang Yin

26 February 2025

Figures 4 and 15 may contain a territory that is disputed according to the United Nations. If and when the manuscript is accepted for final revised publication, you will be asked to choose one of the following options: (a) you could remove the disputed territory from the maps and submit new figure files, or (b) we could add a statement that some figures contain disputed territories.

Answer: Figure 4 has been revised to eliminate political demarcations for enhanced neutrality. For clarity, Figure 15 specifically illustrates localized geographical features without referencing any sovereign entities. Should the need arise, option (b) will be preferentially adopted.